# Experimental Comparison of Laser Cladding and Powder Plasma Transferred Arc Welding Methods for Depositing Wear-Resistant NiSiB + 60% WC Composite on a Structural-Steel Substrate

**DOI:** 10.3390/ma16113912

**Published:** 2023-05-23

**Authors:** Marcin Adamiak, Augustine Nana Sekyi Appiah, Radosław Żelazny, Gilmar Ferreira Batalha, Artur Czupryński

**Affiliations:** 1Materials Research Laboratory, Faculty of Mechanical Engineering, Silesian University of Technology, 18A Konarskiego Street, 44-100 Gliwice, Poland; augustine.appiah@polsl.pl; 2Autoneum Poland Sp. Z o.o., Owsiana 60A, 40-780 Katowice, Poland; radoslawzelazny96@gmail.com; 3Department of Mechatronics and Mechanical Systems Engineering, Polytechnic School of Engineering of the University of Sao Paulo (USP), São Paulo 05508-900, Brazil; gfbatalh@usp.br; 4Welding Department, Faculty of Mechanical Engineering, Silesian University of Technology, 18A Konarskiego Street, 44-100 Gliwice, Poland; artur.czuprynski@polsl.pl

**Keywords:** dilution, heat-affected zone (HAZ), metal matrix composite (MMC), scanning electron microscopy (SEM), transition zone

## Abstract

A Ni-based powder composed of NiSiB + 60% WC was deposited onto a structural-steel substrate using two methods: laser cladding (LC) and plasma powder transferred arc welding (PPTAW). The resulting surface layers were analyzed and compared. Both methods resulted in the precipitation of secondary WC phases in the solidified matrix, but the PPTAW clad exhibited a dendritic microstructure. The microhardness of the clads prepared by both methods was similar, but the PPTAW clad showed higher resistance to abrasive wear compared to the LC clad. The thickness of the transition zone (TZ) was thin for both methods, with a coarse-grain heat-affected zone (CGHAZ) and peninsula-like macrosegregations observed in clads from both methods. The PPTAW clad showed a unique cellular–dendritic growth solidification (CDGS) and a type-II boundary at the TZ attributed to its thermal cycles. While both methods resulted in metallurgical bonding of the clad to the substrate, the LC method exhibited a lower dilution coefficient. The LC method also resulted in a larger HAZ with higher hardness compared to the HAZ of the PPTAW clad. The findings of this study indicate that both methods are promising for antiwear applications due to their wear-resistant properties and metallurgical bonding to the substrate. The PPTAW clad may be particularly useful in applications that require higher resistance to abrasive wear, while the LC method may be advantageous in applications that require lower dilution and larger HAZ.

## 1. Introduction

Surface coatings play a crucial role in enhancing material properties and performance in various industries. For instance, in the rail industry, where structural steel is extensively used in the production of railway components such as wheels, axles, and rails, these coatings are essential due to the harsh operating conditions these components are exposed to, including high loads, abrasion, and corrosion that can lead to premature failure [1,2,3]. Surface coatings have been an integral part of the metals industry for many years, serving to improve the wear and corrosion resistance of components. With advancements in material science and engineering, there has been a rise in surface coating technologies that offer opportunities to enhance the performance and extend the service lives of metallic parts and components [4,5,6]. Studies aimed at improving the wear resistance of metals have explored various forms of coatings, including the use of wires [7] and powders [8]. One approach to coating these materials with wear-resistant powder metal matrix composites is through the use of powders consisting of matrices based on cobalt [9], nickel [10], and other materials reinforced with particles such as WC, TiC, Cr_3_C_2_, VC, BN, TiB_2_, and diamond reinforcements [11,12,13,14,15].

Various methods have been utilized to apply wear-resistant coatings onto steel substrates, including flame spraying, thermal spraying, laser melt injection, and magnetron sputtering [16,17,18], among others. However, these conventional techniques often yield coatings with limited bonding strength and relatively higher porosity, and they are insufficiently thick for heavy-loaded structures such as bearings and gears. In contrast, cladding technology, which encompasses laser cladding (LC) and powder plasma transferred arc welding (PPTAW), presents a cutting-edge approach for coating preparation, enabling complete metallurgical bonding between the coating and substrate [19]. Cladding methods exhibit superior bonding strength, increased coating thickness, and reduced porosity. Moreover, the higher heating and cooling rates associated with cladding can refine the microstructure of the alloys, resulting in enhanced hardness and wear resistance [20,21]. The versatility of LC and PPTAW in working with diverse powders has rendered them popular techniques for surface modification, augmenting strength, hardness, corrosion resistance, and abrasive-wear resistance, as supported by existing literature [22,23,24,25,26].

Tungsten carbide (WC) is widely used in industrial applications due to its excellent wear resistance. However, WC is susceptible to dissolution in the melt pool generated during the cladding process, as it has a low free enthalpy of formation of 38.5 kJ/mol [27]. Therefore, extensive research has been conducted to determine optimal values for the cladding process to minimize carbide dissolution and ensure proper melting. Previous studies have suggested that in the case of LC, laser power ranging from 1.0 kW to 2.0 kW is optimal for cladding meta l= based powder blends containing WC reinforcements [28,29]. Similarly, for PPTAW, studies have reported that plasma gas flow rates (PGFR) ranging from 1.0 L/min to 2.0 L/min yield better cladding performance [30,31].

The performance and properties of a clad are strongly associated with its characteristics. Kim and Lee [32] recently reported that cladding processes, compared to other coating methods, have higher heat inputs and, as a result, lead to an oriented microstructure formation, which has a significant influence on the mechanical properties of the clad. In reported studies regarding Ni-based clads, Yoon et al. [33] and DuPont et al. [34] observed hot cracking. They explained this as resulting from the presence of eutectic phases that were topologically close-packed as a consequence of increased dilution during clad solidification. Additionally, Gittos et al. [35] reported a reduction in the corrosion resistance of their prepared clad after observing interdendritic precipitates and a high content of Fe in the clad. This was reported to have resulted from high dilution. It is therefore useful to apply surface clads using techniques that result in low dilution, such as LC and PPTAW. Such techniques have the potential to mitigate these cladding issues and to produce much more desirable clads.

The transition zone (TZ) at the clad/substrate interface is an area of much microstructural consideration, as the resulting microstructure in this zone impacts the performance of the clad. At the TZ, there is the possibility of hydrogen embrittlement (HE) occurring in the clad. The phenomenon of HE is described by Örnek et al. [36] as the vulnerability of a clad to becoming brittle due to high hardenability, elevated hardness, and the inclusion of partially diluted macrosegregation and a gradient in chemical composition in the TZ. The TZ can also possess a type-II boundary, which is a high-angle grain boundary that is usually in a parallel orientation to the TZ. In the work of Dong et al. [37], it is stipulated that the type-II boundary can serve as an avenue for sulfide stress corrosion cracking (SSCC). In the laser cladding on Ni-steel with stainless steel AISI 304 research conducted by Wu et al. [38], it was reported that the TZ exhibited a type-I boundary and had harder martensite phases. However, during the cladding of this same Ni-steel with a Ni-based alloy, by a different technique, hot-wire gas tungsten arc welding (GTAW-HW), Farias et al. [39] reported the observation of both type-I and type-II boundaries in the TZ without SSCC. It can be inferred from these studies that the resulting microstructure of the TZ and its characteristics vary depending on the cladding method used for material deposition.

Given the increasing use of Ni-based powder blends in diverse industrial applications [40,41,42] and the crucial role of microstructural features in relation to different cladding methods, along with the existing knowledge gap in this area, this study seeks to accomplish two primary goals: (1) to successfully clad a wear-resistant Ni-based powder reinforced with tungsten carbide (WC) with a composition of NiSiB + 60% WC onto a structural-steel substrate using LC and PPTAW techniques, and (2) to conduct a comprehensive comparative analysis of the mechanical properties, such as hardness and wear resistance, as well as the microstructural characteristics of the clads prepared by LC and PPTAW. To achieve these objectives, we used methodologies including scanning electron microscopy (SEM), energy-dispersive X-ray spectroscopy (EDS), light microscopy, digital microscopy, abrasive-wear resistance tests, Vickers microhardness tests, and macroanalytical dilution and penetrant tests. The findings of this research will contribute to knowledge and understanding of these cladding processes and their performance in antiwear applications, filling the existing gap in the literature and providing valuable insights for practical implementation.

## 2. Materials and Methods

### 2.1. Materials and Surface Cladding

#### 2.1.1. Laser Cladding (LC)

The commercially available powder blend, which consists of Ni-Si-B + 60% WC, was acquired from Castolin Eutectic^®^ in Gliwice, Poland. Structural steel grade EN S355 was utilized as the substrate material. The steel plate samples were prepared with dimensions of 90 mm × 30 mm × 10 mm.

The YLS 400 laser system (IPG Photonics, Oxford, MA, USA) was utilized for laser cladding in this study. The process incorporated a powder feeder and direct injection of powder into the melt pool. Argon gas was employed as both the powder transport and shielding gas to protect the clad. The clads were created through 7 multi-runs of single passes with a 50% overlap. Preheating was not employed during the laser processing, and the interpass temperature remained below 30 °C. To determine the optimal power for the clad’s desired performance, the laser power was adjusted between 1.5 kW and 2.0 kW for samples L1 and L2, respectively. All other parameters were kept constant for both samples. The rate of cladding was 0.2 m/min, and the deposition speed was 8 g/min.

#### 2.1.2. Powder Plasma Transferred Arc Welding (PPTAW)

The EuTronic Gap 3511 DC synergic PPTAW system, manufactured by Castolin Eutectic in Gliwice, Poland, was utilized for performing PPTAW. The cladding process involved the use of a plasma torch with a tungsten electrode placed in the center, acting as the cathode to generate a plasma arc, through which powders were fed. The anode was structural-steel-plate substrate material. The plasma gas utilized was argon 5.0 (99.999%) in accordance with ISO 14175-I1:2009 [43], while a shielding and carrier gas mixture of argon/hydrogen 5% H2, Ar (welding mixture ISO 14175-R1-ArH-5 [43]) was employed. The plasma transferred arc had a thermal efficiency coefficient of k = 0.6, with a tungsten electrode (WS-2) measuring 4 mm in diameter and a plasma nozzle size of 3 mm. The plasma gas flow rate (PGFR) was adjusted between 1.0 L/min and 1.2 L/min to prepare samples P1 and P2, respectively. All other PPTAW parameters, such as pilot arc current of 30 A, travel speed of 1.3 mm/s, power flow (cos phi) of 0.99, open-circuit voltage of 95 V (direct current), and plasma transferred arc current of 110 A, remained constant for both samples during the clad deposition.

A summary of the prepared samples is given in Table 1, with their corresponding variation in parameters for both cladding methods.

### 2.2. Characterization

Samples for metallographic analysis were obtained by cutting along the cross-section from the surface clad to the substrate material, as shown in Figure 1. The samples were then subjected to various analyses, including microhardness measurements, scanning electron microscopy (SEM), energy-dispersive X-ray spectroscopy (EDS), light microscopy (LM), and digital microscopy (DM), to further investigate their properties.

The specimens were prepared for metallography by sequentially grinding with SiC papers of various grit sizes including 220, 500, 800, and 1200, followed by polishing with a coarse diamond suspension, and finally mirror polishing using colloidal silica with a particle size of 0.04 μm.

The Vickers microhardness of the surface layers on the cross-section of the clad was evaluated utilizing a 9.81 N load with the FM-ARS 9000 microhardness tester (Future Tech Corporation, Tokyo, Japan). The measurements were carried out subsequent to performing metallographic polishing.

The characterization of powder and evaluation of microstructural features in metallographic samples were conducted utilizing a combination of light microscopy with AxioVision (ZEISS, Jena, Germany), digital microscopy with Leica DVM6 (Leica Microsystems, Heerbrugg, Switzerland), and SEM using Zeiss Evo MA 15 series instrument equipped with an EDS system.

To identify cracks that may have developed in the prepared surface clads after the LC and PPTAW cladding processes, a penetration test was conducted in accordance with the PN-EN ISO 3452 standard [44], utilizing MR^®^ 70 developer, MR^®^ 79 remover (acetone), and MR^®^ 68NF penetrant. Prior to the test, each specimen’s surface was thoroughly cleaned with acetone to remove any impurities. The penetrant was then sprayed onto the surface and left to dry for approximately 10–15 min, after which it was removed using the remover and paper. Next, the developer was sprayed onto the sample’s surface and allowed to settle. Finally, cracks were identified on the surface of the specimen where the penetrant had seeped in.

The prepared surface clads were subjected to tests to determine their resistance to abrasive wear. A reference material, AR400 steel, which is known for its resistance to abrasive wear, was also tested. The testing method used was the “rubber wheel” method, as described in ASTM G65 standard. This method involves placing abrasive materials between the surface being tested and a rubber-lined wheel, as depicted in Figure 2. The rubber-lined wheel is rotated while the surface being tested remains stationary for a certain duration. The testing parameters utilized in the study are detailed in Table 2.

Equations (1)–(3) were used to compute abrasive-wear test parameters. In the equations, M_B_ represents the initial mass of the specimen, M_A_ represents the mass of the specimen after the abrasive-wear test, ρ denotes the density of the material, V_LR_ signifies the average volume loss of the reference material, and V_LS_ denotes the average volume loss of the specimen being studied.
(1)$$\mathrm{Mass}\;\mathrm{Loss},\;\mathrm{ML}\;\left[g\right]=\;\mathrm{MB}\;\left[g\right]\;-\;\mathrm{MA}\;\left[g\right]$$
(2)$$\mathrm{Volume}\;\mathrm{Loss},\;\mathrm{VL}\;\left[{\mathrm{mm}}^{3}\right]=\frac{M_{L}\;\left[g\right]\;}{\rho \;\left[\frac{g}{{\mathrm{cm}}^{3}}\right]}\times 1000$$
(3)$$\mathrm{Relative}\;\mathrm{Abrasive}\;\mathrm{Wear}\;\mathrm{Resistance}\;=\frac{V_{\mathrm{LR}}\left[{\mathrm{mm}}^{3}\right]}{V_{\mathrm{LS}}\left[{\mathrm{mm}}^{3}\right]}$$

## 3. Results and Discussion

### 3.1. Powder Characterization

An SEM image of the NiSiB + 60% WC powder blend used for the cladding processes is shown in Figure 3a. The powder particles showed a mixture of spherical and angular morphologies. In Figure 3b, the result of a laser-assisted powder particle size analysis is presented, with statistical deductions (inset in Figure 3b) revealing a low span (indicating particle uniformity) of 1.3 for the 1475 particles analyzed. This low span value indicates that the powders have a low degree of particle non-uniformity, resulting in improved powder flow during PPTAW and LC processes. In the EDS characterization (Figure 3c–i and Table 3), the spherical powder particles are the Ni-based matrix phase particles, confirmed by the higher content of Ni, relative to other elements. The angular powder particles were found to be WC reinforcements. Boron was too light to be detected by the EDS technique.

The morphology of the powder particles prior to cladding has been reported to affect the performance of the clad for various applications. ArunKumar et al. [45] have reported that angular powder particles perform better in wear-resistant applications, whereas spherical powder particles are more desirable for applications requiring high toughness. According to Li et al. [46], powders that have a mixture of spherical and angular particles result in enhanced stability and function better in applications requiring wear resistance, high hardness, and toughness.

### 3.2. Cladding

A micrograph of the substrate material, EN S355 structural steel grade, was obtained using a light microscope and is presented in Figure 4a. The microstructure mainly consists of allotriomorphic ferrite and pearlite, with traces of Widmanstätten ferrite, which is described as acicular ferrite plates emanating from a prior austenite grain boundary [47]. Microhardness measurements yielded an average of 405 HV with a standard deviation of 6.1 (Figure 4b), and equiaxed grains were observed in the through-thickness direction. The relatively high hardness of this steel grade is attributed to the presence of pearlite and ferrite zones with solute segregation, which can extend up to 400 µm [48], making it suitable for applications requiring high strength and hardness.

Figure 5 shows the top view of the deposited surface clad onto the substrate material. Figure 5a,b are samples prepared using the LC method, and Figure 5c,d are samples prepared using the PPTAW method.

### 3.3. Surface Crack Investigation

The occurrence of surface cracks resulting from the cladding processes was investigated using penetrant testing, and the results are shown in Figure 6. The results reveal that the samples prepared via LC exhibited a significantly higher number of crack sites compared to those prepared via the PPTAW method. This discrepancy can be attributed to the considerably higher heat input from plasma during PPTAW, which promotes better melting and bonding of powder particles, resulting in fewer surface cracks. Sample P2, prepared with a PGFR of 1.2 L/min in the PPTAW method, demonstrated better resistance to surface crack formation compared to sample P1, prepared with a PGFR of 1.0 L/min. This observation has been previously explained by the entrapment of plasma gas in the plasma jet at lower PGFR, and an increase in the PGFR leads to increased heat in the system and expansion of the plasma jet, which fosters melting of powder particles and reduces pore formation and surface cracks [10].

On the other hand, it was noted that the number of crack sites relatively increased with an increase in laser power from 1.5 kW to 2.0 kW in the LC method. This phenomenon has been attributed to the differences in thermal expansion coefficients of powder particles and the higher hardness of WC, resulting in weaker interfacial bonding between powder particles at higher laser powers, leading to increased surface crack sites [49]. Sample L1, prepared with a laser power of 1.5 kW in the LC method, showed better resistance to surface crack formation compared to sample L2, prepared with a laser power of 2.0 kW.

It is worth noting that cladding surface cracks do not significantly impact the overall wear performance of the clad, as reported by earlier research [50]. In some cases, the crack sites can even be utilized as lubrication sites for applications that require surface lubricants [51].

### 3.4. Microhardness Tests

The results of room-temperature Vicker’s microhardness measurements of the prepared clads are visualized in the plot in Figure 7. It is observed that the mean hardness of the matrix phase in the clad prepared by the LC method is relatively higher than the mean of the matrix phase in the clad prepared by the PPTAW method. For the LC samples, L1 had matrix mean hardness of 695 HV and sample L2 had a mean of 634 HV. Samples P1 and P2 from the PPTAW method had matrix mean hardnesses of 600 HV and 620 HV respectively. Point measurements taken from the reinforcing carbide phases had contrasting mean values; PPTAW samples had higher values than LC samples. For LC samples, L1 had a carbide mean value of 2139 HV and L2 had a mean of 2120 HV. Samples P1 and P2 had average values of 2244 HV and 2261 HV, respectively. The standard deviations, as indicated by the bars in the plot, suggest that the mean values are not significantly further from each other. This outcome is expected because both processes are relatively high-energy-density processes, and the powder used is the same. However, it is observed in this hardness testing that sample L1 from LC showed higher hardness than sample L2. Although higher laser power introduces higher energy input in the cladding process, it is reported that increasing the laser power can also result in turbulent flow, resulting in reduced particle melting and a corresponding reduction in clad hardness [52]. It is also observed that sample P2 from the PPTAW method showed higher hardness than sample P1. As the PGFR was increased, the corresponding rise in plasma heat input led to higher particle temperatures, accelerating the dissolution of carbides into the matrix. This resulted in improved adhesion between the particles, leading to lower porosity and enhanced hardness, as reported by [11] in a similar PPTAW study.

### 3.5. Abrasive-Wear Resistance Tests

To assess the abrasion resistance of the surface clads in metal-mineral abrasive-wear conditions, tests were conducted using Hardox 400 (AR400) as a reference material, which is a known abrasion-resistant material widely used in previous research [53,54,55,56]. The volume loss was determined by considering the density differences between WC, metallic matrix, and the reference material. The results of the abrasive-wear tests are presented in Table 4.

The findings reveal that the abrasive-wear performance of the surface clads varied depending on the process parameters for both LC and PPTAW methods. Specifically, the LC sample L1, prepared with a laser power of 1.5 kW, exhibited higher relative abrasive wear compared to sample L2, which had a laser power of 2.0 kW. Additionally, sample P2, fabricated with a PGFR of 1.2 L/min, showed superior relative abrasive-wear resistance compared to sample P1, which had a PGFR of 1.0 L/min.

The findings from the macroanalysis of surface cracks, microhardness measurement, and abrasive-wear resistance tests revealed appreciable results for sample L1 from the LC method and sample P2 from the PPTAW method. These samples showed superior performance compared to their counterparts that were prepared with variations in the process parameters. As a result, these select samples were chosen for further in-depth studies and investigations to gain a deeper understanding of the wear mechanism, microstructural evolution, and dilution mechanisms of the clads prepared by both methods in a comprehensive comparative analysis.

Digital micrographs in Figure 8 depict the surface of the clads before and after undergoing tests to determine their resistance to abrasive wear. The mechanism underlying the wear resistance was investigated and found to occur in two distinct stages. Initially, as the rubber wheel was set in motion with quartz sand in contact with the surface of the layer, the surface underwent wear. This process continued until the quartz sand had removed enough material, bringing the wear debris, which was rich in tungsten carbide (WC), closer to the quartz sand. When these WC particles came into contact with the quartz sand, smearing of the WC particles was observed. With increasing contact time, the WC particles reduced friction between the quartz sand and the revolution of the rubber wheel, resulting in a significant decrease in wear [57]. Micrographs in Figure 8 reveal that WC particles were exposed at the surface when further abrasion was hindered, and regions on the surface without WC particles (matrix) showed deeper abrasion compared to regions with a higher concentration of WC particles.

### 3.6. Microstructural Evolution

The metallographic cross-section of the clad and substrate material was examined in Figure 9 for samples fabricated using LC and PPTAW methods. The cross-section revealed distinct zones, including the clad subsurface, middle zone (MZ), transition zone (TZ), and heat-affected zone (HAZ). Microscopic analysis of the clad in both samples displayed varying degrees of particle dispersion, believed to be WC. Further magnification of the MZ in both samples obtained using a light microscope is shown in Figure 10.

Upon solidification of the deposited clads, particle size analysis revealed different characteristics of WC particles in both the LC clad and the PPTAW clad. In the LC clad, the measurements showed a minimum size of 20 µm, a maximum size of 216 µm, and a mean particle size of 167.2 µm, with a standard deviation of 41.6. On the other hand, the PPTAW clad exhibited a minimum carbide particle size of 16 µm, a maximum size of 229 µm, and a mean size of 126.8 µm, with a standard deviation of 55.2. The mode particle sizes were determined to be 189 µm and 196 µm for the LC clad and PPTAW clad, respectively. These findings are further supported by the images in Figure 9, where it is evident that the carbide particles in the PPTAW clad appear larger compared to those in the LC clad. This disparity suggests that the PPTAW method resulted in a more even distribution of carbide particles within the clad, in contrast to the LC method. The discrepancy in particle size between the two methods can be attributed to the operational conditions. When employing the LC method, as described by Kotarska et al. [52], a laser power of 1.5 kW resulted in turbulent flow during the powder feeding process. This turbulence led to collisions among carbide particles, causing them to break down and resulting in relatively smaller sizes upon solidification of the clad. In contrast, the PPTAW method operated at a powder gas flow rate (PGFR) of 1.2 L/min, ensuring a consistent flow of powder particles. This continuous flow facilitated a more uniform dispersion of carbides within the clad, leading to the observed larger particle sizes in the PPTAW clad compared to the LC clad.

#### 3.6.1. Middle Zone (MZ)

Figure 11 shows an SEM image of the MZ in the laser-clad sample, accompanied by EDS maps illustrating the distribution of elements across the examined MZ area. The EDS maps reveal non-uniform dispersion of primary WC particles within the Ni-based matrix, with precipitation of secondary WC phases observed, as shown in the inset in Figure 12. The MZ of the PPTAW sample, as observed under SEM, is presented in Figure 13, along with EDS maps illustrating the distribution of powder constituents in the MZ after solidification. The matrix exhibits a dendritic multi-directional microstructure upon solidification, characterized by the presence of partially dissolved primary carbides and precipitates of secondary carbides dispersed throughout.

EDS maps from Figure 11 and Figure 13 confirm that the bright-grey regions in the SEM images correspond to the primary carbides, and the analyzed precipitates shown in Figure 12 and Figure 14 are the secondary precipitates nucleated from the WC reinforcing phase during solidification. The EDS maps reveal consistent segregation of Si and W in the clads produced by both methods. The heat input required to fully melt the Ni-based matrix (with a melting point of 1555 °C) surpasses the melting temperature of Si (1410 °C). Consequently, the Si phases undergo melting prior to other materials. Due to the strong tendency to form a WSi_2_ phase, the liquid Si forms bonds with W, leading to their observed segregation at the same locations according to EDS analysis. Döscher et al. [58] reported the formation of a WSi_2_ phase at processing temperatures below 1414 °C, resulting from a solid-state reaction between W and Si. However, when processing temperatures exceed 1414 °C, the WS_2_ phase forms from a reaction between molten Si and solid W. In typical MMC clads, ceramic reinforcement particles tend to agglomerate upon solidification due to density differences between the carbides and the matrix [59]. However, in the samples prepared in this study using LC and PPTAW methods, agglomeration of carbides was not observed, which can be attributed to the effective mixing of powder particles, resulting in the dispersion of carbides in the matrix phase.

During the cladding process, partial dissolution of the carbide particles (with a melting point of 2785 °C) in the completely dissolving Ni-based matrix phase is responsible for the observation of the dendritic structure formed in the MZ. Earlier research [60] has indicated that the bonding of these partially dissolving carbides in the matrix improves the stability of the clad, which is enhanced even further by the precipitation of secondary phases via accelerated diffusion.

#### 3.6.2. Transition Zone (TZ)

Figure 15a provides an overview of the transition zone (TZ) for sample L1. Upon closer examination of the TZ (Figure 15b), the presence of precipitates with the same composition as those identified in the middle zone (MZ) can be observed, indicating nucleation of precipitates at the TZ. The heat-affected zone (HAZ) shown in Figure 15c is characterized by coarse grains composed of martensite laths and can be described as a coarse-grain heat-affected zone (CGHAZ). Additionally, a peninsula-like macrosegregation is observed at the TZ, as depicted in Figure 15d.

Similarly, the overview of the TZ for sample P2 is shown in Figure 16a. The TZ displays distinct regions, as shown in Figure 16b: a planar solidification growth region and a cellular–dendritic solidification growth (CDSG) region. The HAZ for this sample (Figure 16c) also exhibits characteristics of a CGHAZ, like that observed in sample L1. A type-II boundary is present at the TZ of sample P2, as shown in Figure 16d. Furthermore, peninsula-like macrosegregation is also observed at the TZ of sample P2, as depicted in Figure 16e. A magnified image of the CDSG region containing interdendritic precipitates is shown in Figure 16f.

The SEM images reveal that the transition zones in the samples prepared using LC and PPTAW methods are remarkably thin, measuring approximately 9 µm for LC and approximately 11 µm for PPTAW. Importantly, the TZs in both methods appear free from any discontinuities, such as partially dissolving powder particles, hot cracking, or absence of fusion. Additionally, the substrates of both LC and PPTAW samples exhibit clear evidence of metallurgical bonding with the clad material. In sample L1, the TZ shows a melt pool of the matrix phase that terminates in a planar mode upon solidification, which is a commonly observed phenomenon in cladding processes with relatively slower cooling rates, such as LC. However, the higher thermal cycles (heating and cooling) of the PPTAW method, which are approximately 10^8^ K/s [61], and the associated high heat absorption by the substrate material, relative to the LC method, which has thermal cycles of approximately 10^4^ K/s [62], resulted in a different solidification mode. The solidification advanced in three stages, starting with a planar-mode solidification. This was then followed by a cellular-mode solidification and terminated with a cellular–dendritic-mode solidification. The observed transitions in the solidification mode are linked to the solid–liquid interface instability. This is caused by a decrease in the rate of cooling, as the substrate also becomes heated during the cladding process. In addition, constitutional supercooling, which is described as the reduction in the solid solubility relative to the liquid [63], accounts for the observed changes in solidification mode. The PPTAW clad had a higher rate of nucleation close to the TZ, resulting from the substrate’s heat sink and the presence of several sites for nucleation. This observation is consistent with the findings of Xibao et al. [64], who reported that some of these sites for nucleation emanate from partially melted powder particles.

The occurrence of type-II boundaries at the transition zone (TZ) was observed exclusively in plasma powder transferred arc welding (PPTAW), while laser cladding (LC) did not exhibit this phenomenon. Previous research by Nelson et al. [65] has proposed that the transformation of ferrite (δ-γ) allotropic phases in the coarse-grain heat-affected zone (CGHAZ) during cooling is a critical prerequisite for the formation of type-II boundaries in structural steel. Recently, Farias et al. [39] postulated that the interface between the substrate and the clad is mobile and can induce projection of the CGHAZ into the clad during cooling, resulting in the formation of type-II boundaries. It is worth mentioning that the type-II boundary was only observed in a small section of the TZ. This is because the precipitation occurring in the TZ during the transition in the solidification mode impedes the advancement of the type-II boundary in the planar growth region [25].

Both PPTAW and LC samples exhibited tiny peninsula-like macrosegregations in the TZ of the clad, as observed in Figure 15d and Figure 16e. The stagnation of the substrate liquid layer, which is very thin, accounted for this observation. It can be deduced from this that the dilution was low, resulting from the disparity in the substrate and MMC fusion temperatures, as well as the turbulence generated by the laser power and the PGFR. Consequently, the melted substrate was transported into the clad pool, where it solidified rapidly due to the presence of adjacent melted powder (liquid). According to the mechanism of the formation of type-I macrosegregation as proposed by Yang et al. [66], this type of macrosegregation limits the inter-liquid chemical diffusion. This observed phenomenon is an indication that LC and PPTAW used for these cladding operations are low-dilution processes and are able to impede the formation of macrosegregations. The absence or smaller size of macrosegregations is advantageous for improved clad performance. This is because, as proposed by Soysal et al. [67], the macrosegregations can potentially function as sites for hydrogen embrittlement because of their higher hardness.

The growth mode of cellular–dendritic solidification, as observed in the PPTAW sample, is a result of clad-pool grain-growth competition. This means that grains that possess higher growth rates impede the growth of their neighboring grains. This leads to the formation of an oriented microstructure. It is noteworthy that the fine interdendritic precipitates in the sample were not continuous, and no hot cracks were detected, indicating that the MMC clad and the structural-steel substrate exhibit good weldability. Furthermore, the PPTAW process, which has a higher cooling rate compared to LC, resulted in a greater presence of interdendritic precipitates.

Figure 17 depicts the microanalysis of the TZ in the LC sample using EDS. Similarly, Figure 18 presents the microanalysis of the TZ in the PPTAW sample. The EDS maps in these figures reveal the presence of a chemical composition gradient along the TZ, involving Si, Ni, W, C, and Fe, which can be attributed to the lower dilution during the cladding process. The gradient is not as steep as in mechanical bonding, indicating that the substrate and clad are metallurgically bonded through a diffusion layer. Additionally, the distribution of Fe content in close proximity to the TZ is found to be heterogeneous, with a decrease in Fe content observed with increasing distance from the TZ.

### 3.7. Dilution and Heat-Affected Zone (HAZ)

Figure 19 illustrates the relationship between the dilution ratio (D) and the area of dilution, as calculated using Equation 4. The reinforcement area (Fn) and the fusion zone area (Fw) were determined based on the geometrical schematic shown in Figure 19a. To obtain these values, measurements were taken using the AutoCAD software program (Autodesk, CA, USA) for the height of the clad on the substrate material (R), penetration depth of the clad into the substrate (P), and width of the clad across the x-plane (w). The results of the measurements and the computed values of the dilution coefficient D for the investigated clads are summarized in Table 5.
(4)$$\mathrm{Dilution},D=\mathrm{Fw}/\left(\mathrm{Fw}+\mathrm{Fn}\right)\times 100$$

The findings indicate that LC demonstrated a significantly lower dilution ratio of 2.1% in comparison to PPTAW, which had a dilution ratio of 4.5%. These results are in line with previous literature that has consistently reported dilution ratios of less than 10% for both LC and PPTAW, contrasting with other surface-cladding technologies such as metal arc welding, gas tungsten arc welding, and gas metal arc welding, which typically exhibit dilution ratios ranging from 10% to 30%. Thus, the findings of this study support the existing literature’s observations of low dilution ratios for LC and PPTAW [68,69].

In Figure 20, the microhardness measurement results of the heat-affected zone (HAZ) are presented, accompanied by a digital micrograph depicting the HAZ. The microhardness values of both samples L1 and P2 in this region were found to be higher than that of the substrate material. Notably, the LC sample exhibited a larger HAZ of approximately 283 µm (Figure 20a) compared to the PPTAW sample, with approximate HAZ of 35 µm (Figure 20c), attributed to the relatively higher heat retention of the LC process. The average microhardness value of the LC sample was measured to be 680 HV (standard deviation of 17.0), while the PPTAW sample had an average microhardness of 480 HV (standard deviation of 5.6). The HAZ in both samples predominantly consisted of martensite laths, forming what is commonly referred to as the coarse-grain heat-affected zone (CGHAZ), which is consistent with previous studies by Barick et al. [70] on ferritic steel. Furthermore, the increase in microhardness of the HAZ in both samples compared to the substrate material aligns with findings from earlier studies [71,72], which also reported higher microhardness in the coarse-grain structure of the HAZ.

## 4. Conclusions

A wear-resistant powder blend of composition NiSiB + 60% WC was successfully deposited on a structural-steel substrate using LC and PPTAW methods. There was precipitation of secondary WC phases in the matrix upon solidification for clads prepared by both methods. However, the clad from PPTAW had a dendritic microstructure, which is attributed to its relatively higher thermal cycles than those of the LC process.Penetrant test macroanalysis revealed that the clad prepared by the LC method had more surface cracks than the PPTAW counterpart, owing to the lower laser heat input. The microhardness of the matrix and reinforcing WC phases of the clads prepared by both methods were comparatively similar. However, when compared to a reference abrasive-wear-resistant material, AR400, the relative abrasive-wear resistance of the PPTAW clad was relatively higher, at 4.7, than the clad prepared by LC, at 4.5. The wear mechanism was found to be the same for clads prepared by both methods.The TZs for both clads were observed to be thin (approx. 9 µm for LC and approx. 11 µm for PPTAW), with a coarse-grain heat-affected zone (CGHAZ), made up of martensite laths, and a peninsula-like macrosegregation observed for clads from both methods. However, the PPTAW clad had a cellular–dendritic growth solidification (CDGS) and a type-II boundary at the TZ, which is explained as resulting from the thermal cycles of this method.The LC clad had a lower dilution coefficient, at 2.1%, than the PPTAW clad, with dilution coefficient of 4.5%. The higher heat retention of the LC method resulted in a larger HAZ (≈283 µm) with higher hardness (average of 680 HV) than the HAZ of the PPTAW clad, which was ≈35 µm and had an average hardness of 480 HV. EDS analysis showed elemental diffusion at the clad/substrate interface with a diffusion gradient, which shows that the bonding of the clad by both methods was metallurgical, making them more desirable for industrial antiwear applications.

## Figures and Tables

**Figure 1 materials-16-03912-f001:**
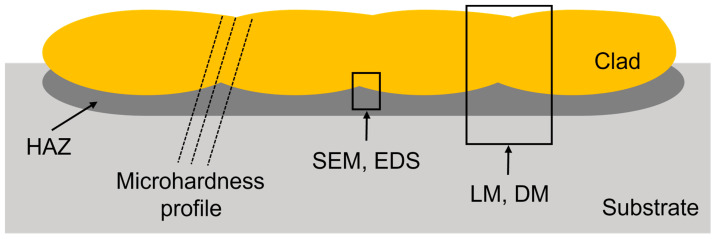
Cross-section of metallographic sample showing the areas examined via microhardness testing, SEM, EDS, LM, and DM.

**Figure 2 materials-16-03912-f002:**
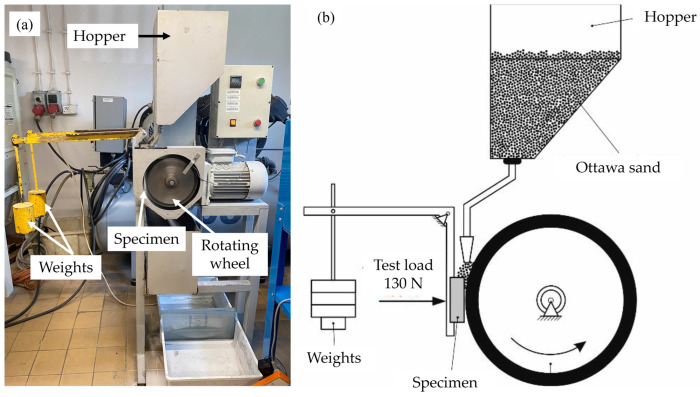
(**a**) Abrasive-wear resistance testing setup based on ASTM G65, (**b**) schematic diagram of the process.

**Figure 3 materials-16-03912-f003:**
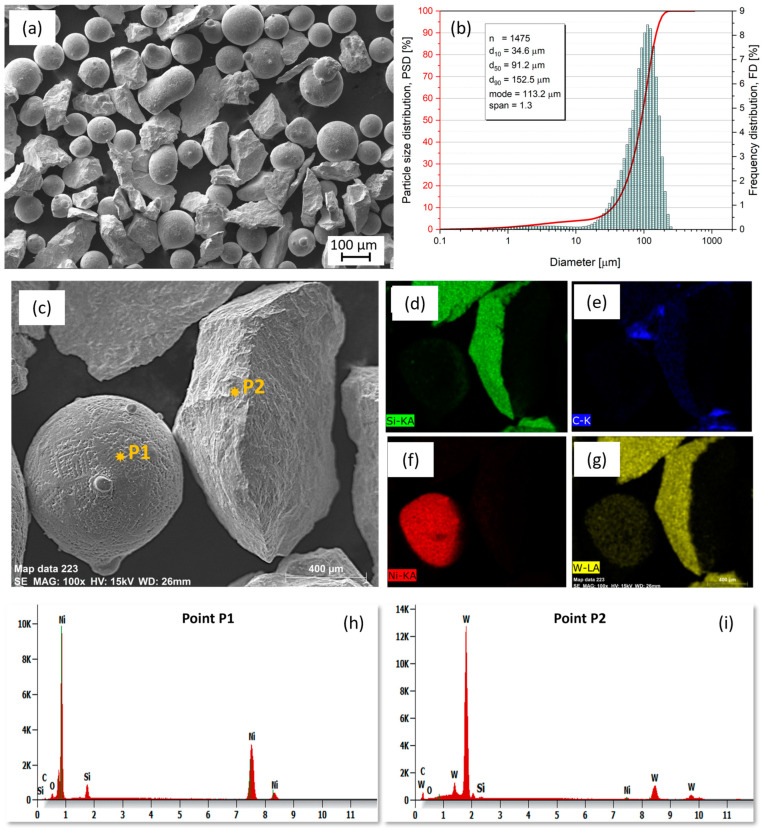
(**a**) NiSiB + 60% WC powder-blend particles as observed under the scanning electron microscope, (**b**) powder-blend particle size distribution (PSD) plot showing statistical parameters and powder particle span, (**c**–**g**) EDS elemental maps showing the position and amounts of elements present in the analyzed powder particles, (**h**,**i**) X-radiation energy diagrams of measured points from analyzed powder particles.

**Figure 4 materials-16-03912-f004:**
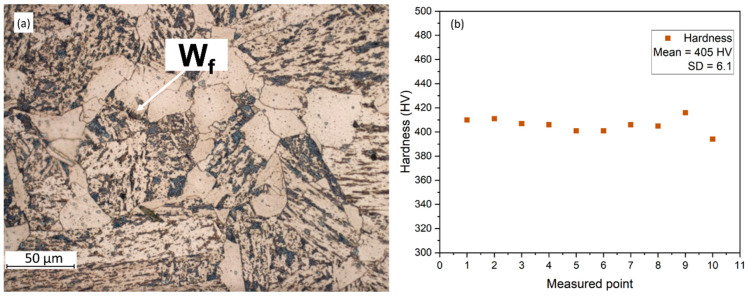
(**a**) Microstructure of substrate material, steel grade EN S335 showing the site of Widmanstätten ferrite, (**b**) microhardness (HV) of the base material.

**Figure 5 materials-16-03912-f005:**
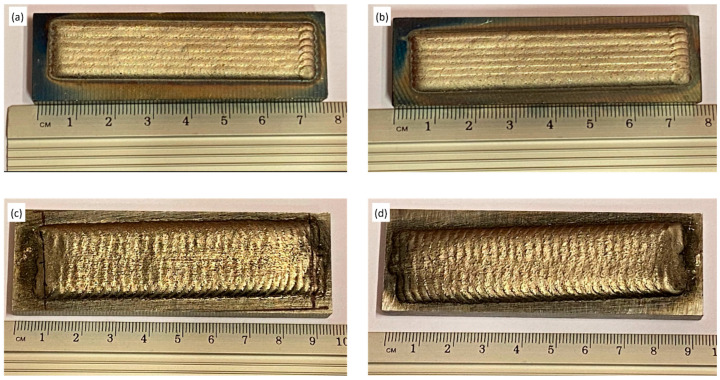
Top view of samples after cladding: (**a**,**b**) samples L1 and L2 prepared by laser cladding, (**c**,**d**) samples P1 and P2 prepared by PPTAW.

**Figure 6 materials-16-03912-f006:**
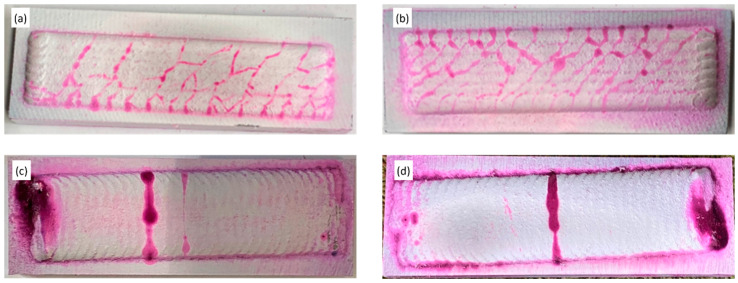
Top view of samples after penetrant test: (**a**,**b**) samples L1 and L2 prepared by laser cladding, (**c**,**d**) samples P1 and P2 prepared by PPTAW.

**Figure 7 materials-16-03912-f007:**
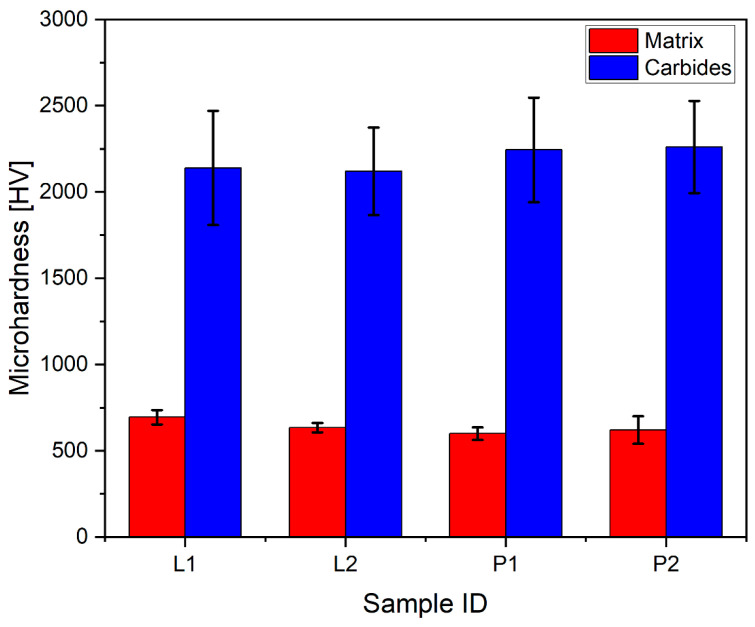
Average microhardness results of the matrix and carbide phases of the clads prepared by LC and PPTAW.

**Figure 8 materials-16-03912-f008:**
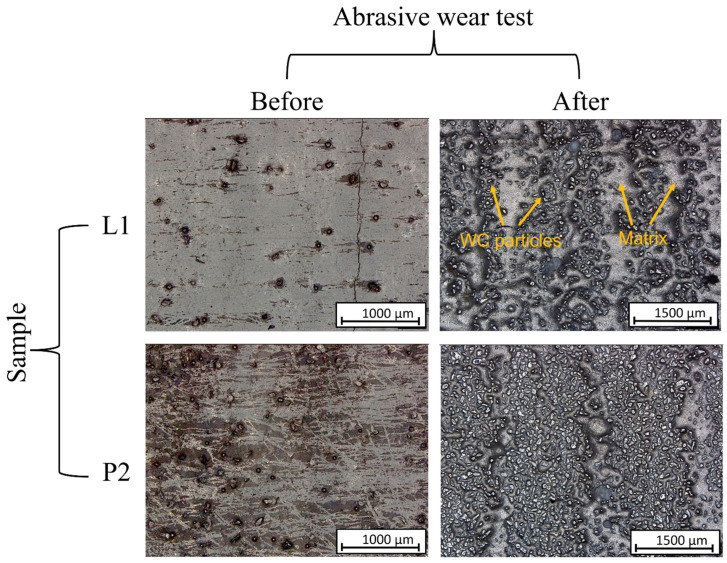
Digital micrographs of surface clads before and after abrasive-wear resistance tests for sample prepared by laser cladding (L1) and sample prepared by PPTAW (P2).

**Figure 9 materials-16-03912-f009:**
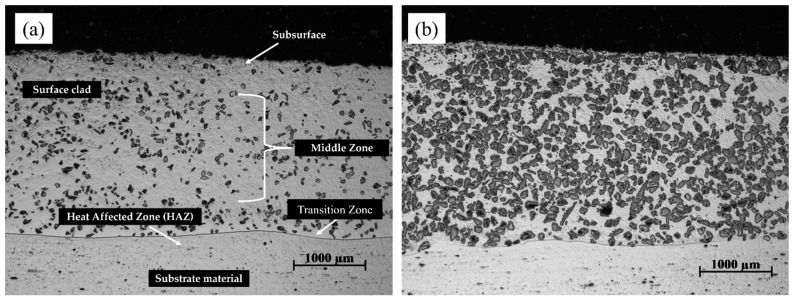
Light micrographs of prepared surface clads showing the observable zones across the cross-section: (**a**) sample L1 prepared by laser cladding, (**b**) sample P2 prepared by PPTAW.

**Figure 10 materials-16-03912-f010:**
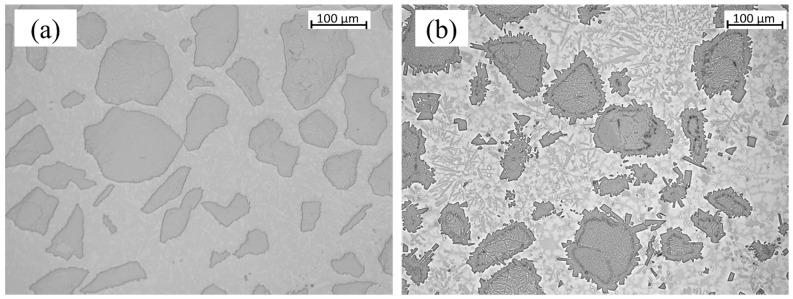
Light micrograph of the middle zone of surface clads at 200× magnification: (**a**) sample L1, (**b**) sample P2.

**Figure 11 materials-16-03912-f011:**
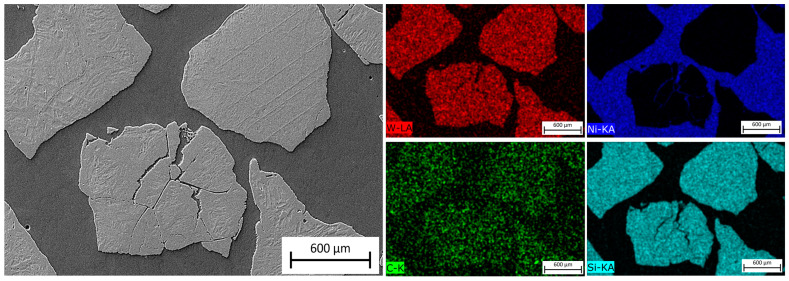
SEM image and maps from EDS microanalysis of the middle zone of sample L1 showing the distribution of the MMC clad constituent within the matrix.

**Figure 12 materials-16-03912-f012:**
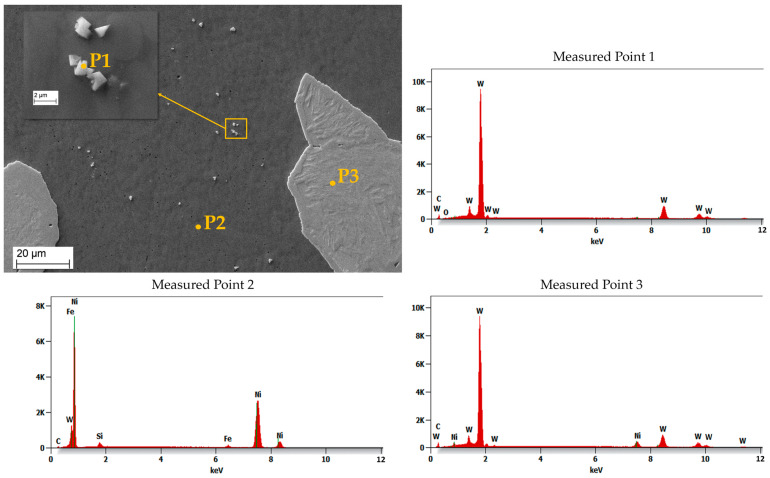
Energy-dispersive X-ray spectroscopy (EDS) microanalysis of the middle zone of surface clad for sample L1 prepared by laser cladding.

**Figure 13 materials-16-03912-f013:**
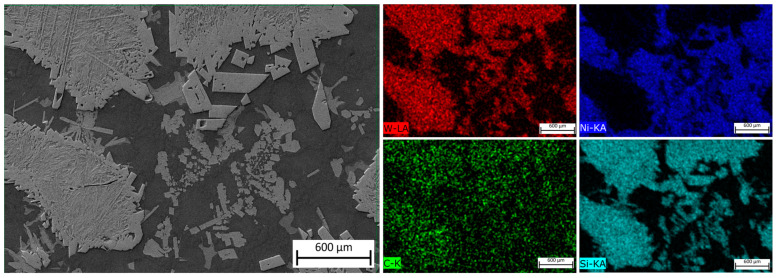
SEM image and maps from EDS microanalysis of the middle zone of sample P2 showing the distribution of the MMC clad constituent within the matrix.

**Figure 14 materials-16-03912-f014:**
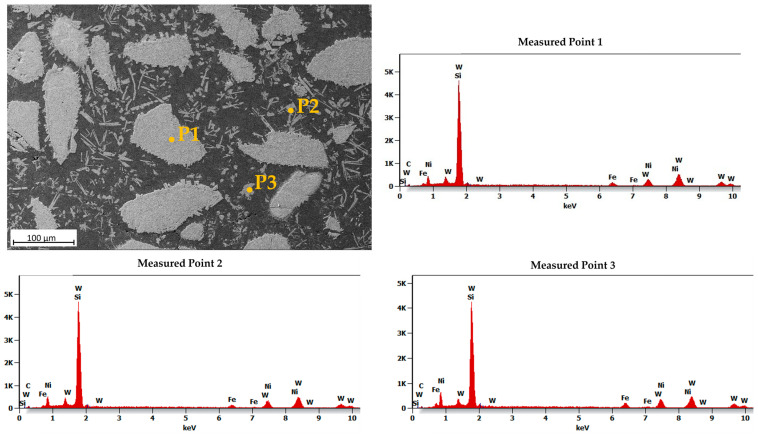
Energy-dispersive X-ray spectroscopy (EDS) microanalysis of the middle zone of surface clad for sample P2 prepared by PPTAW, showing the precipitates of secondary WC particles in the matrix.

**Figure 15 materials-16-03912-f015:**
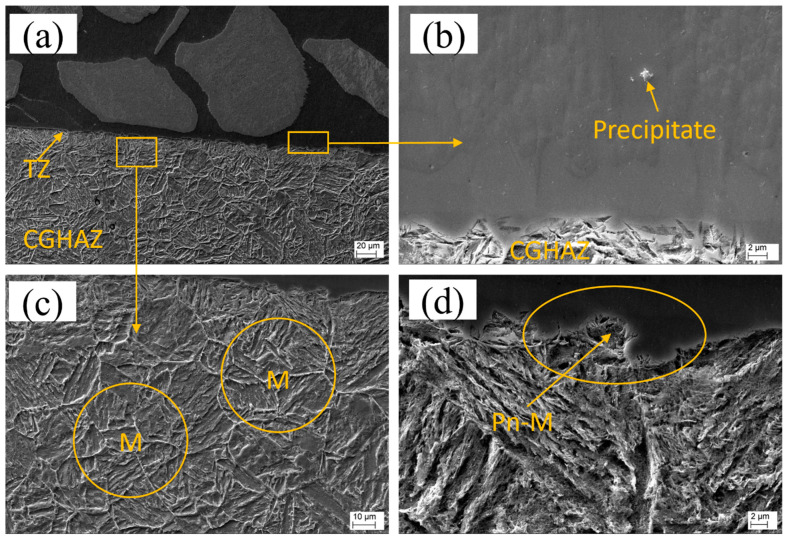
Scanning electron micrographs of sample L1: (**a**) overall view of the transition zone (TZ) showing the coarse-grain heat-affected zone (CGHAZ), (**b**) precipitation in the TZ, (**c**) CGHAZ martensite laths (M) in the substrate, (**d**) peninsula-like macrosegregation (Pn-M).

**Figure 16 materials-16-03912-f016:**
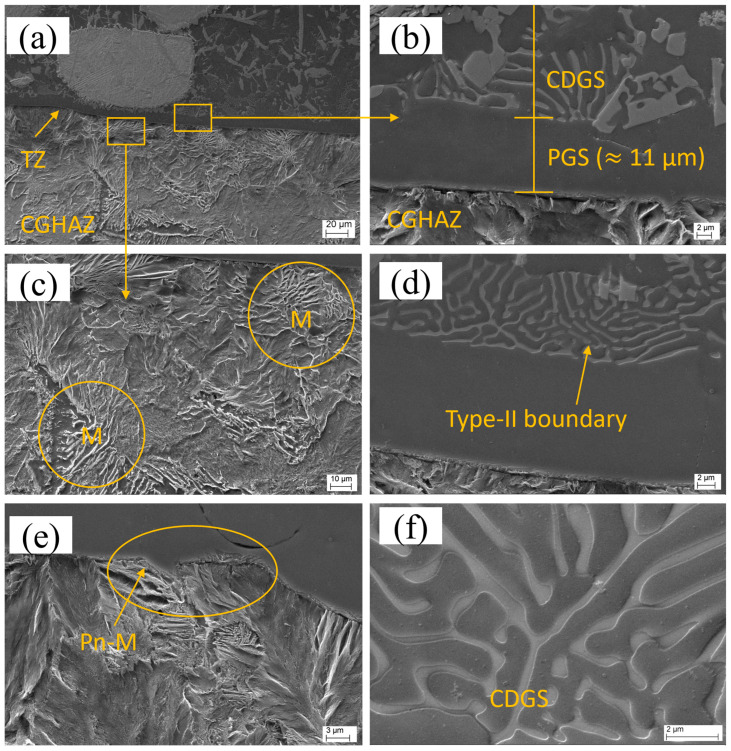
Scanning electron micrographs of PPTAW sample P2: (**a**) overall view of the transition zone (TZ) showing the coarse-grain heat-affected zone (CGHAZ), (**b**) planar growth solidification (PGS) and cellular–dendritic growth solidification (CDGS), (**c**) CGHAZ with martensite laths (M) in the substrate, (**d**) type-II boundary, (**e**) peninsula-like macrosegregation, (Pn-M), (**f**) oriented cellular/dendritic growth at the TZ containing interdendritic precipitates.

**Figure 17 materials-16-03912-f017:**
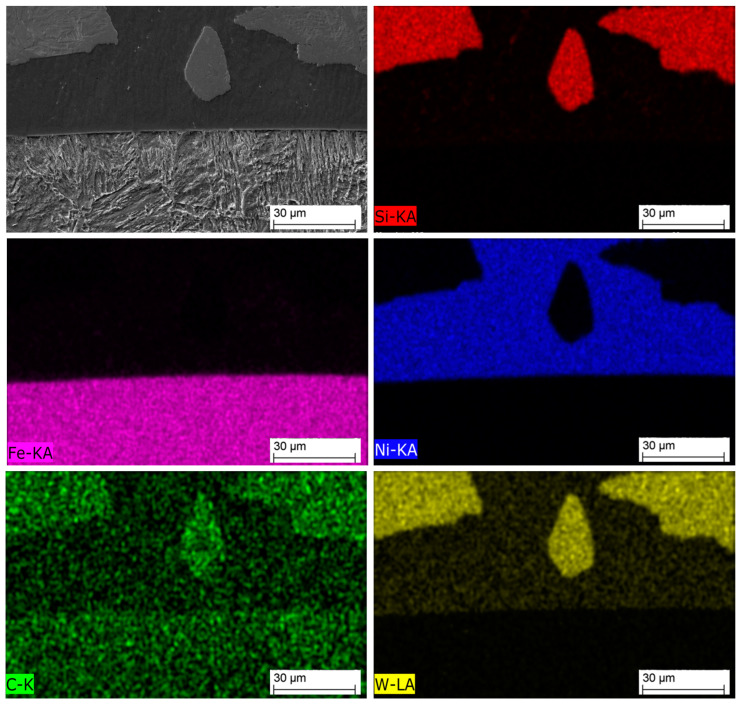
Energy-dispersive X-ray spectrometry mapping of the transition zone of laser-cladded sample L1.

**Figure 18 materials-16-03912-f018:**
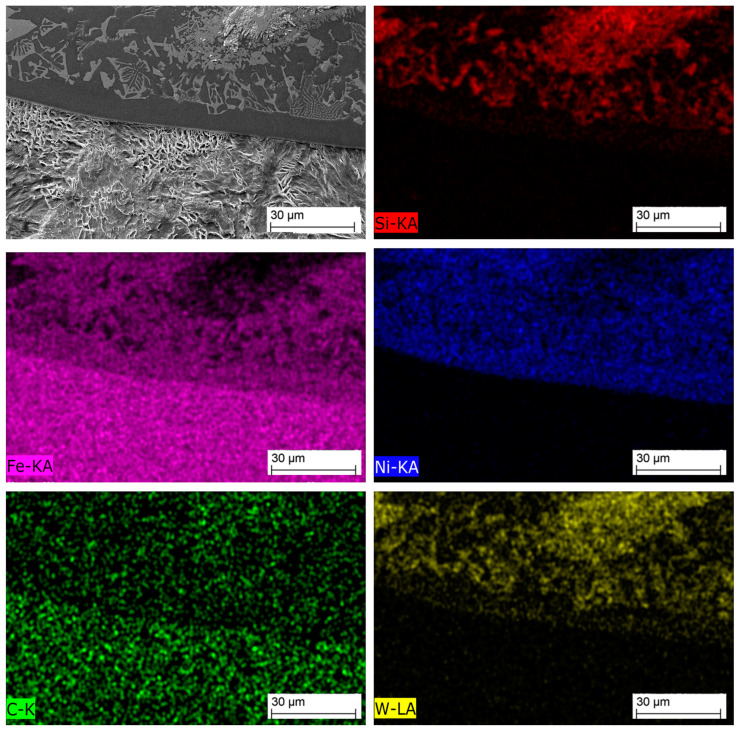
Energy-dispersive X-ray spectrometry mapping of the transition zone of PPTAW sample L1.

**Figure 19 materials-16-03912-f019:**
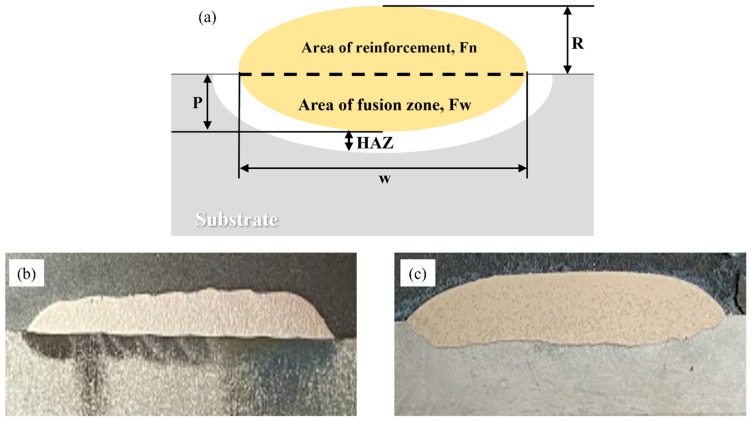
(**a**) Schematic representation of the geometric parameters of the cross-section of the final material after deposition of the clad onto the substrate material, depicting the area of reinforcement (Fn), area of fusion (Fw), heat-affected zone (HAZ), clad layer height on the substrate material (R), depth of clad penetration into the substrate material (P), and width of the clad across the x-plane (w). (**b**) Cross-sectional view of sample L1. (**c**) Cross-sectional view of sample P2.

**Figure 20 materials-16-03912-f020:**
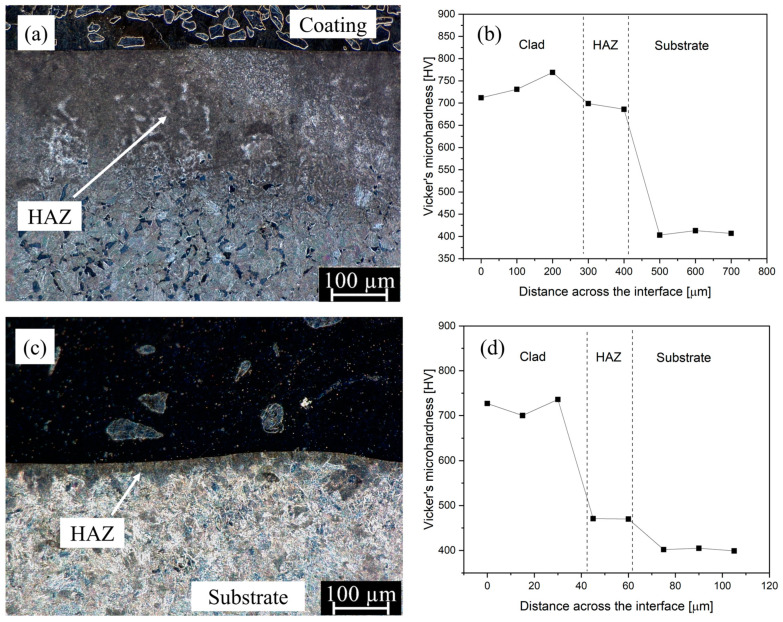
Digital micrograph of the cross-section of investigated sample showing the coating/cladding area, the heat-affected zone (HAZ) and the substrate area (Left), and microhardness profile of points taken along the indicated cross-section (Right). (**a**,**b**) Sample L1, (**c**,**d**) Sample P2.

**Table 1 materials-16-03912-t001:** Prepared samples with their respective parameter variations.

Sample ID	Cladding Method	Laser Power [kW]	PGFR * [L/min]
L1	LC *	1.5	–
L2	LC	2.0	–
P1	PPTAW *	–	1.0
P2	PPTAW	–	1.2

* LC–Laser cladding; PPTAW–Powder plasma transferred arc welding: PGFR–Plasma gas flow rate.

**Table 2 materials-16-03912-t002:** Abrasive-wear resistance testing parameters.

Parameter	Value	Unit
Abrasive particle grain size	210–297	µm
Feed rate	335	g/min
Pressure	130	Pa
Rubber wheel turns	6000	turns
Test time	30	min

**Table 3 materials-16-03912-t003:** Chemical composition in Weight % and Atom % of measured points from analyzed powder particles.

		C	O	Si	Ni	W
Point P1	Atom %	6.6	1.7	6.3	84.9	–
	Weigh t%	1.5	0.5	3.4	94.4	–
Point P2	Atom %	54.4	3.1	0.3	0.3	41.8
	Weight %	7.7	0.6	0.4	0.2	91.1

**Table 4 materials-16-03912-t004:** Results of abrasive-wear resistance tests of surface clads composed of NiSiB + 60% WC prepared by laser cladding and powder plasma transferred arc welding methods in comparison to the abrasive-wear resistance of AR400 reference material.

Sample	Mass before Test, [g]	Mass after Test, [g]	Mass Loss, [g]	Material Density, [g/cm^3^]	Volume Loss, [mm^3^]	Relative Abrasive-Wear Resistance
Surface clads						
L1	159.5765	159.2499	0.3266	11.1935	29.1776	4.5
L2	165.1587	164.6972	0.4615	11.1935	41.2292	3.2
P1	196.0594	195.6905	0.3689	11.1935	32.9566	4.0
P2	195.6418	195.3264	0.3154	11.1935	28.1771	4.7
Reference Samples						
H1	104.6219	103.4971	1.0318	7.7836	132.5607	1.0
H2	111.7377	110.7989				

**Table 5 materials-16-03912-t005:** Geometrical properties and dilution ratio of prepared surface clads.

Sample	Cladding Method	Layer Height, R (mm)	Penetration Depth, P (mm)	Layer Width, w (mm)	Dilution, D, %
L1	LC	1.8	0.2	16	2.1
P2	PPTAW	2.7	0.5	24	4.5

## Data Availability

Not applicable.

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
