# Peer review of "Experimental Comparison of Laser Cladding and Powder Plasma Transferred Arc Welding Methods for Depositing Wear-Resistant NiSiB + 60% WC Composite on a Structural-Steel Substrate"

_materials, 2023, doi:10.3390/ma16113912_

Round 1

Reviewer 1 Report

In my opinion this manuscript can be accepted in the present form. The investigated topic is interesting. A wide characterization is carried out and the results support the aims.

Author Response

Remark: In my opinion this manuscript can be accepted in the present form. The investigated topic is interesting. A wide characterization is carried out and the results support the aims.

Response: The authors are grateful for your acknowledgment of their scientific work to put this manuscript together and appreciate your acceptance of this manuscript for publication in this reputable journal.

Reviewer 2 Report

The current manuscript is a good and interesting work. However, the finding showed no significant difference in the results, but its importance is to demonstrate the fabrication of MMC with two closed methods. In my opinion, I accept the manuscript after the minor revisions

1-      The samples code (L1, L2, P1 and P2 ) should be listed with their abbreviations in separate table

2-      Fig.11 should be replaced with a high-resolution image, the scale bar should be clear, and the SEM image notes under each image should be removed.

3-      Fig. 11 and 13  EDX images for (W) and (Si) elements are the same; please check

4-      In Fig. 20 , the test point in the hardness curve , you should mention the test point unit ( mm. or micro ), then replace and modify the figure after updating it.

The article's written language is adequate, and there are no severe mistakes.

Author Response

Remark: The current manuscript is a good and interesting work. However, the finding showed no significant difference in the results, but its importance is to demonstrate the fabrication of MMC with two closed methods. In my opinion, I accept the manuscript after the minor revisions.

  1. Comment: The samples code (L1, L2, P1 and P2) should be listed with their abbreviations in separate table

Response: A table (Table 1) has been added to section 2 that lists the sample IDs and their associated process parameter variations.

  1. Comment:11 should be replaced with a high-resolution image, the scale bar should be clear, and the SEM image notes under each image should be removed.

Response: The image in Figure 11 has now been replaced with a high-resolution image, the scale bar has been made more legible, and the SEM image notes under each image has been removed. Figures 13, 17 and 18 have also been revised to meet these requirements.

  1. Comment: 11 and 13 EDX images for (W) and (Si) elements are the same; please check

Response: In the revised version of the manuscript, a discussion of this observation with relevant literature, has been included in section 3.6.1. This appears in the manuscript as follows.

“The EDS maps reveal consistent segregation of Si and W in the clads produced by both methods. The heat input required to fully melt the Ni-based matrix (with a melting point of 1555 oC) surpasses the melting temperature of Si (1410 oC). Consequently, the Si phases undergo melting prior to other materials. Due to the strong tendency to form a WSi2 phase, the liquid Si forms bonds with W, leading to their observed segregation at the same locations according to EDS analysis. Döscher et al. [57] reported the formation of a WSi2 phase at processing temperatures below 1414 oC, resulting from a solid-state reaction between W and Si. However, when processing temperatures exceed 1414 oC, the WSi2 phase forms from a reaction between molten Si and solid W.”

  1. Comment: In Fig. 20, the test point in the hardness curve, you should mention the test point unit (mm. or micro), then replace and modify the figure after updating it.

Response: “Test point” has been replaced with “Distance across the interface”

 (µm)”. Figure 20 has therefore been updated and replaced.

Reviewer 3 Report

The paper is devoted to the experimental comparative investigation and evaluation of prospective of Laser Cladding and Powder Plasma Transferred Arc Welding for the formation of composite WC-containing deposits on a structural steel.

The topic is relevant and can be of interest for the wide scientific community involved in thermal spraying, welding as well as composite materials development.

The complex approach including comparison of data from the different research techniques seems to be really beneficial and gives a clearer insight into the deposition processes as well as structure-properties relationships.

The paper contains enough details, appropriate citations and coherent conclusions, although I found few points to be addressed:

I consider the term “High(er)-thermal Cycles” used in the paper as not-completely appropriate and even perplexing (what is high – temperature, velocity, etc.?) and recommend to define it more precisely.

One more comment concerns the abbreviation “MMC”. As can be seen from the Figure 3, the powder used in the research is not actually composite. I would propose the terms “powder mixture” or “powder blend” instead.  The abbreviation “MMC” can be only used for the deposits description.

The essentially higher size and content of carbides in the P2 sample as compared with the L1 one (see Figure 9) as well as the absence of silicon in the Ni-rich areas and presence of Si in WC particles (see Figures 11 and 13) has not been mentioned and explained in the text, although it is important from the materials science point of view (remember the title of the journal) and can affect significantly the materials corrosion and mechanical performance.

Taking into account the clear-cut practical orientation on the rail industry in the introduction, it would be nice to give more specific practical recommendations in the conclusions.

Finally, some misprints have to be corrected in the text, i.e.:

Page 2, line 50: Cr3C2 is likely thought about,

MZ is used as the abbreviation for the middle zone in Page 13, line 358 and others, while it is referred to as the matrix zone in Page 15, line 401

Author Response

Remark: The paper is devoted to the experimental comparative investigation and evaluation of prospective of Laser Cladding and Powder Plasma Transferred Arc Welding for the formation of composite WC-containing deposits on a structural steel.

The topic is relevant and can be of interest for the wide scientific community involved in thermal spraying, welding as well as composite materials development.

The complex approach including comparison of data from the different research techniques seems to be really beneficial and gives a clearer insight into the deposition processes as well as structure-properties relationships.

The paper contains enough details, appropriate citations and coherent conclusions, although I found few points to be addressed:

  1. Comment: I consider the term “High(er)-thermal Cycles” used in the paper as not-completely appropriate and even perplexing (what is high – temperature, velocity, etc.?) and recommend to define it more precisely.

Response: The manuscript has been revised to remove or replace the word “higher” which appeared anywhere in the manuscript and used to make unjustified qualifications. In the case of thermal cycles, which is a crucial concept in this research, more precise values have been indicated in the first instance of its appearance in the manuscript, accompanied by relevant citations in section 3.6.2. Subsequently, all other instances where the authors indicated the word “higher thermal cycles” are in reference to the first instance.

  1. Comment: One more comment concerns the abbreviation “MMC”. As can be seen from the Figure 3, the powder used in the research is not actually composite. I would propose the terms “powder mixture” or “powder blend” instead. The abbreviation “MMC” can be only used for the description of the deposit.

Response: Throughout the revised manuscript, the “powder blend” has been used to replace and describe the powders used for the cladding processes. The abbreviation “MMC” is now used exclusively to describe the deposits.

  1. Comment: The essentially higher size and content of carbides in the P2 sample as compared with the L1 one (see Figure 9) as well as the absence of silicon in the Ni-rich areas and presence of Si in WC particles (see Figures 11 and 13) has not been mentioned and explained in the text, although it is important from the materials science point of view (remember the title of the journal) and can affect significantly the materials corrosion and mechanical performance.

Response: The authors have included discussions with relevant literature to address this omission.

To address the carbide particle disparities after the clad deposition, the authors have included the following in section 3.6.

“Upon solidification of the deposited clads, particle size analysis revealed different characteristics of WC particles in both the LC clad and the PPTAW clad. In the LC clad, the measurements showed a minimum size of 20 µm, a maximum size of 216 µm, and a mean particle size of 167.2 µm, with a standard deviation of 41.6. On the other hand, the PPTAW clad exhibited a minimum carbide particle size of 16 µm, a maximum size of 229 µm, and a mean size of 126.8 µm, with a standard deviation of 55.2. The mode particle sizes were determined to be 189 µm and 196 µm for the LC clad and PPTAW clad, respectively. These findings were further supported by the images in Figure 9, where it is evident that the carbide particles in the PPTAW clad appear larger compared to those in the LC clad. This disparity suggests that the PPTAW method resulted in a more even distribution of carbide particles within the clad, in contrast to the LC method. The discrepancy in particle size between the two methods can be attributed to the operational conditions. When employing the LC method, as described by Kotarska et al. [50], a laser power of 1.5 kW resulted in turbulent flow during the powder feeding process. This turbulence led to collisions among carbide particles, causing them to break down and resulting in relatively smaller sizes upon solidification of the clad. In contrast, the PPTAW method operated at a powder gas flow rate (PGFR) of 1.2 L/min, ensuring a consistent flow of powder particles. This continuous flow facilitated a more uniform dispersion of carbides within the clad, leading to the observed larger particle sizes in the PPTAW clad compared to the LC clad.”

To address the observation of the absence of Si in the matrix and its presence in the reinforcing phase, the authors have added the following discussion to section 3.6.1

The EDS maps reveal a consistent segregation of Si and W in the clads produced by both methods. The heat input required to fully melt the Ni-based matrix (with a melting point of 1555 oC) surpasses the melting temperature of Si (1410 oC). Consequently, the Si phases undergo melting prior to other materials. Due to the strong tendency to form a WSi2 phase, the liquid Si forms bonds with W, leading to their observed segregation at the same locations according to EDS analysis. Döscher et al. [57] reported the formation of a WSi2 phase at processing temperatures below 1414 oC, resulting from a solid-state reaction between W and Si. However, when processing temperatures exceed 1414 oC, the WSi2 phase forms from a reaction between molten Si and solid W.”

  1. Comment: Taking into account the clear-cut practical orientation on the rail industry in the introduction, it would be nice to give more specific practical recommendations in the conclusions.

Response: The authors acknowledge this observation regarding the application of the results of this study. However, the rail industry was used as an example of industries where wear resistance is mostly needed or applied. To reflect this intention, the authors have introduced the phrase “For instance, in the rail industry…” in the introduction, on line 2. All other mentions of the rail industry in the manuscript are representative of all industrial applications requiring wear resistance.

  1. Comment: Finally, some misprints have to be corrected in the text, i.e.:

Page 2, line 50: Cr3C2 is likely thought about,

Response: The manuscript has now been checked by the authors for misprints. The identified misprint in the comment has been corrected, on line 49, in the current form of the manuscript.

  1. Comment: MZ is used as the abbreviation for the middle zone in Page 13, line 358 and others, while it is referred to as the matrix zone in Page 15, line 401

Response: The abbreviation MZ has been used throughout the manuscript as the middle zone in the current form of the manuscript. All misprints regarding this have been revised.
